# Synthesis, Electrochemical Studies, and Antimicrobial Properties of Fe_3_O_4_ Nanoparticles from *Callistemon viminalis* Plant Extracts

**DOI:** 10.3390/ma13214894

**Published:** 2020-10-31

**Authors:** Gloria E. Uwaya, Omolola E. Fayemi, El-Sayed M. Sherif, Harri Junaedi, Eno E. Ebenso

**Affiliations:** 1Department of Chemistry, Faculty of Natural and Agricultural Sciences, North-West University (Mafikeng Campus), Private Bag X2046, Mmabatho 2735, South Africa; okonyeglo@yahoo.com (G.E.U.); Eno.Ebenso@nwu.ac.za (E.E.E.); 2Material Science Innovation and Modelling (MaSIM) Research Focus Area, Faculty of Natural and Agricultural Sciences, North-West University (Mafikeng Campus), Private Bag X2046, Mmabatho 2735, South Africa; 3Center of Excellence for Research in Engineering Materials (CEREM), King Saud University, P.O. Box 800, Al-Riyadh 11421, Saudi Arabia; esherif@ksu.edu.sa; 4Department of Physical Chemistry, Electrochemistry and Corrosion Laboratory, National Research Centre, El-Buhouth St., Dokki, Cairo 12622, Egypt; 5Mechanical Engineering Department, College of Engineering, King Saud University, P.O. Box 800, Al-Riyadh 11421, Saudi Arabia; hjunaedi@ksu.edu.sa

**Keywords:** green-mediated, Fe_3_O_4_-NP, *Callistemon viminalis*, characterization, cyclic voltammetry, antimicrobial activity

## Abstract

Less toxic, environmentally safe green-mediated iron (III) oxide nanoparticles (Fe_3_O_4_-NP) synthesized using *Callistemon viminalis* (*C. viminalis*) leaf (Fe_3_O_4_-NPL) and flower (Fe_3_O_4_-NPF) extracts is reported in this work for the first time. Total flavonoids and phenols present in the plant extracts were determined. Characterization of the nanoparticles was carried out using Fourier transform infrared (FTIR) spectroscopy, ultraviolet–visible spectroscopy (UV–VIS), scanning electron microscopy (SEM), transmission electron microscopy (TEM), X-ray diffraction (XRD), and Malvern zeta sizer. Other properties of the nanoparticles were investigated using the thermogravimetric analyser and cyclic voltammetry. The average particle sizes obtained for Fe_3_O_4_-NPL and Fe_3_O_4_-NPF were 17.91 nm and 27.93 nm, respectively. Fe_3_O_4_-NPL exhibited an excellent electrochemical activity when compared with Fe_3_O_4_-NPF based on a stability study using cyclic voltammetry and regression value. Additionally, Fe_3_O_4_-NPF displayed excellent antimicrobial activity against *Bacillus cereus*, *Salmonella enteritidis*, and *Vibrio cholerae* with zones of inhibition of 13, 15, and 25 mm, respectively. Simple, cheap, and less toxic green-mediated iron (III) oxide nanoparticles synthesized from *C. viminalis* leaf (Fe_3_O_4_-NPL) and flower (Fe_3_O_4_-NPF) extracts hold the potential of being used to control the activity of pathogenic bacteria of health importance and as an electrochemical sensor for both biological and environmental analytes.

## 1. Introduction

Iron oxide nanoparticles have gained scientific interest due to their unique super-paramagnetic properties [1,2], large surface-area-to-volume ratio [1,2], ease of preparation [1], affordability, and less toxicity [2]. It has found application in water treatment [3], examination and treatment of cancerous cells [4], and impedance of resistance of drugs to human blood chronic myelogenous leukaemia K562 cells [5,6]. Nanosized metal oxide particles have been used as electrochemical sensors for the detection of various analytes such as serotonin in physiological samples like urine [7], arsenic (III), and lead (II) in water [8] due to their extraordinary optical, electrical, and magnetic properties [9]. 

The methods used to synthesize Fe_3_O_4_-NP include hydrothermal [10], solvothermal [11], precipitation [12], and biological methods, which include the use of fungi, bacteria, algae, and plants (green) [13,14]. Attention has been directed to the green method of synthesizing nanoparticles owing to its low toxicity. Therefore, green synthesis of Fe_3_O_4_-NP has been reported using the leaves of *Hordeum vulgare* and *Rumex acetosa*, peel extracts of *Punica* (pomegranate) with 0.05 M FeSO_4_ [15], and seed and leave extracts of *Lagenaria siceraria* with FeCl_3_.6H_2_O as the precursor [13,16]. 

*Callistemon viminalis* (*C. viminalis*) is a woody shrub plant with a pleasant smell commonly called the weeping brush [17]. *C. viminalis* is of the family Myrtaceae [17] with vast medical and biological values, which include antibacterial, fungal, anticancer, anti-inflammatory, and anthelmintic activities [17,18]. In the area of scientific research, *C. viminalis* (leaf and flower extracts) has been reported to be used in the synthesis of nanoparticles, such as gold nanoparticles (Au-NP), samarium (III) oxide (Sm_2_O_3_), silver dimer (Ag_2_-NP), and chromate (III) oxide (α-Cr_2_O_3_) [17,19,20,21], because they are rich in polyphenols and flavonoids [17]. *C. viminalis* extracts act as a reducing agent, reducing iron (III) ions (Fe^3+^) to zero valence metal. Leaf and flower extracts and essential oils obtained from *C. viminalis* leaves provide antioxidant and antibacterial activities against some selected bacterial strains, such as *Pseudomonas aeruginosa*, *Serratia marcescens*, *Escherichia coli*, *Bacillus cereus*, *Sarcina lutea*, *Bacillus subtilis*, and *Salmonella typhi* [22].

This study reports the synthesis of iron (III) oxide nanoparticles (Fe_3_O_4_-NP) using *Callistemon viminalis* (*C. viminalis*) leaf and flower extracts. Total phenolic content (TPC) was determined for both the leaf and flower extracts. An electrochemical study of Fe_3_O_4_-NP on screen-printed carbon electrode (SPCE-Fe_3_O_4_-NP) was also evaluated to establish the potential applicability of Fe_3_O_4_-NP as an electrode modifier for an electrochemical sensor. In addition, the study affirmed the antimicrobial properties of Fe_3_O_4_-NP against pathogens of health significance. The synthesized Fe_3_O_4_-NP were characterized by using Fourier transform infrared (FTIR) spectroscopy, ultraviolet–visible spectroscopy (UV–VIS), scanning electron microscopy (SEM), transmission electron microscopy (TEM), energy-dispersive X-ray spectroscopy (EDS), X-ray diffraction (XRD), thermogravimetry (TGA), and zeta sizer. The electron transport properties of the nanoparticles at a modified screen-printed carbon electrode (SPCE) were investigated via electrochemical studies by using cyclic voltammetry. 

## 2. Experimental Section 

### 2.1. Materials and Equipment 

*Fresh Callistemon viminalis* (*C. viminalis*) leaves and flowers were harvested from herbarium of the North-West University, Mafikeng, South Africa. Reagents used include; iron (III) chloride hexahydrate with 99% purity, 5 mM potassium hexacyanoferrate (III) (K_3_Fe(CN)_6_) solution in 0.1 M phosphate buffer solution (PBS) with pH 7.4 prepared using 0.1 M sodium hydrogen phosphate dihydrate (NaH_2_PO_4_.2H_2_O) and disodium hydrogen phosphate dihydrate (Na_2_HPO_4_.2H_2_O), dimethylformamide (CH_3_)_2_NC(O)H), methanol, 10% aluminium III chloride (AlCl_3_) solution, 7.5% sodium trioxocarbonate (IV) solution (Na_2_CO_3_), 5% sodium nitrite (NaNO_3_), 5% sodium hydroxide solution, gallic acid, quercetin, 10% Folin–Ciocalteu reagent (FCR), and distilled water. Malvern zeta sizer instrument version 7.11 was used for zeta potential measurement. Electrochemical studies were conducted using metrohm DropView software version 200 and 4 mm disposable screen-printed electrodes fitted with DropSens potentiostat. Thermal stability and UV-Vis characterization of Fe_3_O_4_-NP were examined by SDT Q600 V20.9 Build 20 thermogravimetric analyser and Uvline 9400 SCHOTT instrument produced in EU accordingly. 

### 2.2. Preparation of C. viminalis Leaf and Flower Extracts

Harvested leaves and flowers of *C. viminalis* plant were washed thoroughly with tap water, rinsed with distilled water, and dried under sunlight for 48 h. About 5 g of the ground dried leaves was weighed and boiled with 100 mL distilled water for 15 min at 80 °C [13]. The leaf extract was allowed to cool under room temperature and filtered using a vacuum pump with a Büchner funnel and 125 mm Whatman filter paper. The colour of the leaf extract was an orange with 5.32 pH. The same procedure was followed to obtain a red-coloured flower extract, which gave a 3.89 pH value.

### 2.3. Phytochemical Test of Extracts 

To determine the presence of flavonoids and phenolic compounds in the *C. viminalis* leaf and flower extracts, a preliminary test was conducted [23]. The presence of flavonoids in the extracts of the plant was determined using an alkaline reagent test with 2 mL of the extracts with four drops of 10% aqueous sodium hydroxide solution (10 g of NaOH in 90 g of distilled water) plus a lead acetate test of 0.1 M with 2 mL of the extracts. The presence of phenolic compounds in the extracts was determined by a ferric test using 0.1 M of alcoholic iron (III) chloride hexahydrate [23].

### 2.4. Total Determination of Phenol Content 

An established TPC method with modification was employed in estimating the phenol content of the *C. viminalis* leaf and flower extracts using gallic acid as an internal standard as reported in the literature [24]. Five milligrams of dried ground *C. viminalis* leaves and flowers was dissolved in 5 mL methanol. Ten percent FCR and 6% Na_2_CO_3_ were prepared by the addition of water in a 100 mL volumetric flask. An amount of 300 μL of each extract was taken into a glass test tube followed by the addition of 1.5 mL of 10% FCR solution. The mixture was sonicated for 5 min. Thereafter, 1.5 mL of 6% Na_2_CO_3_ solution was added, and the test tubes were incubated in a dark place for 2 h at room temperature. Gallic acid standard at various concentrations (750, 650, 550, 450, 350, and 150 μg/mL) was prepared in the extracts simultaneously as described in the established protocol [24]. The *C. viminalis* extracts and gallic acid standard solutions were measured at 760 nm with a UV–visible spectrophotometer after a blank measurement (methanol). The phenol content of the extracts was estimated using the gallic acid standard calibration curve, and the total phenols were expressed as a microgram of gallic acid equivalent (GAEmg) per 5 g of dry extract.

### 2.5. Total Determination of Flavonoid Content 

The extracts were prepared following the same procedure in phenol content. Five percent NaNO_3_, 5% NaOH, and 7% AlCl_3_ solutions were prepared by using water in a 100 mL volumetric flask. The leaf and flower (300 μL) extracts were taken into a glass test tube; 75 μL of 5% NaNO_3_ was thereafter added and left for 5 min. Then 1.25 mL of AlCl_3_ and 0.5 mL of NaOH were added. The mixture was sonicated for 5 min and incubated for 5 min at room temperature [24]. Concurrently, quercetin standard of different concentrations (500, 300, 100, 50, and 25 μg/mL) was also prepared. The absorbance of all the working solutions and standard solutions was measured at 510 nm against blank methanol [24].

### 2.6. Synthesis of Fe_3_O_4_-NP from Extracts

Fe_3_O_4_-NP was synthesized from the *C. viminalis* leaf extract (Fe_3_O_4_-NPL). Iron (III) chloride hexahydrate (10 mM) solution (99% purity) was added to the leaf extract in a 1:1 volume ratio. A change in colour from yellow to black was observed, indicating the formation of Fe_3_O_4_-NPL. The mixture was stirred for 3 h and allowed to stand at room temperature for 1 h to obtain colloidal suspension. Colloidal suspension was thereafter centrifuged at 25 °C, 6000 rpm, and an acceleration of 9 m s^−2^ for 30 min and washed several times with ethanol and dried in an oven at 40 °C for 2 h. Fe_3_O_4_-NP from the *C. viminalis* flower extract (Fe_3_O_4_-NPF) was synthesized following the same steps as in the Fe_3_O_4_-NPL synthesis [16]. 

### 2.7. Characterization of Iron (III) Oxide Nanoparticles 

Synthesized iron (III) oxide nanoparticles from the *C. viminalis* leaf and flower extracts (Fe_3_O_4_-NPL and Fe_3_O_4_-NPF) were characterized with the use of ultraviolet–visible spectroscopy, energy-dispersive spectroscopy (EDS), Fourier transform infrared (FTIR) spectroscopy, scanning electron microscopy (SEM), transmission electron microscopy (TEM), X-ray diffraction (XRD), thermogravimetric analysis (TGA), and Malvern zeta sizer.

### 2.8. Electrochemical Characterization 

An electrochemical study was carried out by using cyclic voltammetry. The screen-printed carbon electrode (SPCE) was modified with Fe_3_O_4_-NPF and Fe_3_O_4_-NPL paste (2 mg of Fe_3_O_4_-NP with 3 drops of dimethylformamide sonicated for 24 h). A DropView 2000 software potentiostat was used, and a quercetin solution of 10 mM potassium hexacyanoferrate (III) (K_3_[Fe(CN)_6_]) in 0.1 M PBS as an electrolyte aimed at the efficient transport of ionic charges across the electrode with potential set at −0.2 V.

### 2.9. Antimicrobial Studies 

Considering the properties and applications of Fe_3_O_4_-NP, it can be concluded that Fe_3_O_4_-NP hold the potential of inhibiting the replication of pathogens (antimicrobial agents) [25]. The antimicrobial properties of the green-mediated Fe_3_O_4_-NP were investigated by using agar disk diffusion methods as outlined by CLSI (2011) [26]. The antimicrobial activity of the nanoparticles was examined against eight categories of bacterial pathogens of public health importance belonging to both Gram-positive and Gram-negative types. Bacterial pathogens include *Bacillus cereus* (ATCC 10876) as Gram-positive, *E. coli* (ATCC 259622), *Enterococcus faecalis* (ATCC 29212), *Enterococcus faecium* (ATCC 700425), *Enterococcus gallinarum* (ATCC 701221), *Salmonella typhimurium* (ATCC 14028), *Salmonella enteritidis*, (ATCC 13076), and *Vibrio cholerae*. Preferred bacterial pathogens were reactivated in Luria–Bertani broth overnight before use, and their concentrations adjusted to 10^6 CFU/mL. The antimicrobial disk used was sterilized in an autoclave at 121 °C for 15 min at 15 pascals. An antimicrobial disk was impregnated in dispersed nanoparticles of different concentrations (100 and 150 µg/mL). Mueller–Hinton agar (MHA) was prepared according to the manufacturer’s design and was sterilized. MHA was poured into sterile petri dishes and made to solidify before its seeding with 10^6 CFU/mL of chosen pathogens. To check mate overlapping, the nanoparticle-impregnated antimicrobial disk was placed in equidistance to each other while the antibiotic, ciprofloxacin (30 µg), was used as control. Plates were aerobically incubated for 24 h at 37 °C, after which zones of inhibition were observed and measured using a graduated meter rule in millimetres.

## 3. Results and Discussion

### 3.1. Phytochemical Test 

On treating the leaf and flower extracts of *Callistemon viminalis* (*C. viminalis*) with the alkaline reagent of sodium hydroxide, brown- and an orange-coloured-precipitates were formed with yellow- and blue-coloured precipitates, respectively, when the leaf and flower extracts were treated with 0.1 M lead acetate solution, indicating the presence of flavonoids. Grey-black and black solutions were formed on adding a few drops of 0.1 M alcoholic iron (III) chloride hexahydrate on the flower and leaf extracts, indicating the presence of phenols.

### 3.2. Determination of Total Phenolic Content (TPC) in Callistemon viminalis

The TPCs of the *C. viminalis* leaf and flower extracts estimated from the calibration curve of gallic acid standard (Figure 1) using the Folin–Ciocalteu assay at 760 nm were 12.636 and 29.320 GAEmg/g, respectively, based on Equation (1),
(1)$$C=\frac{cV}{m}$$
where *C* mg/ml is the total phenolic content in gallic acid equivalent (GAE), *c* is the concentration of gallic acid established from the calibration curve, *V* is the volume of extract, and *m* is the mass of extracts used. The calibration curves complied with the Beer–Lambert law with 0.9998, 0.002 μg/mL, and 0.1066 regression, slope, and intercept values accordingly.

### 3.3. Determination of Total Flavonoid Content (TFC) in Callistemon viminalis

The total flavonoid compositions of the *C. viminalis* leaf and flower extracts from the quercetin standard calibration curve as presented in Figure 2 were 19.371 and 0.634 QEmg/g using Equation (1). QE is total flavonoid content in quercetin equivalent. The content of flavonoids was higher in the *C. viminalis* leaf extract than in the flower extract. The calibration curves obeyed the Beer–Lambert law. The regression, slope, and intercept values obtained from the curve were 0.9647, 0.0007 and 0.5701 μg/mL, respectively. Comparing the TFCs of the leaf and flower extracts of *C. viminalis*, the content of phenol in the flower extract was greater than in the leaf contract, which is in agreement with the literature [27]. 

The higher TFC in the leaves than in the flowers was supposedly due to the abundant shikimic acid in the leaves compared with those in other organs, while the lower TFC in the flowers was indicative of the depletion of flavonoids due to colour development in the flower petals. Shikimic acid is a precursor in the flavonoid biosynthetic pathway and has been reported to be higher in leaves than in other organs due to the photosynthetic activities in the leaves [27,28]. 

The higher TPC in the flowers in this study could have been influenced by the developmental phase. The associated colour change of the petals due to oxidation resulting in the easy rupture and release of the cell wall bonded phenolic compounds in the flowers. The higher TPC in this study corroborates the report on groundcover rose flowers [20]. An increase in total phenol content was reported to be influenced by the stages of development and colour change [29]. 

### 3.4. Spectroscopic and Morphological Characterization

Fourier transform infrared (FTIR) analysis was carried out on the extracts and synthesized Fe_3_O_4_-NP in order to determine the compounds and the bonding properties present. In the FTIR spectra (Figure 3a), the leaf and flower extract (Figure 3b) spectrum displayed important vibration peaks at about 3300, 1729, and 1724, which are assigned to the O-H stretch of the alcohol/phenol group and C = O of the carbonyl group (carboxylic acid, aldehyde, ketone) accordingly. These peaks are possible reducing agents that are responsible for the formation of iron (III) oxide nanoparticles. The observed peak at 1735 in FeCl_3_.6H_2_O is ascribed to the deformation of water molecules, suggesting the existence of physio sorbed water on the surface of ferric chloride [30]. The FTIR spectrum of Fe_3_O_4_-NP synthesized by the *C. viminalis* leaf extract (Fe_3_O_4_-NPL) showed a reduced intensity in the vibration peaks at 3205 and 1735 corresponding to the O-H stretch of the alcohol/phenol group and the O-H stretch of the carboxylic acid group, aldehyde, and ketone (Figure 3a), respectively. On the other hand, the peak at 508 and 578 cm^−1^ were the Fe-O stretching of Fe_3_O_4_-NP (Figure 3a). For Fe_3_O_4_-NP synthesized by the *C. viminalis* flower extract (Fe_3_O_4_-NPF), as shown in Figure 3b, the peaks corresponding to the presence of the alcohol/phenol group (3209) and carbonyl group (1727) were identified, while the peaks at 600 and 520 cm^−1^ corresponded to the bending and Fe-O stretching of Fe_3_O_4_-NPF, respectively. The asymmetry and symmetry stretching vibration of COO^−^ (carboxylate group) that proved the combination of protein with nanoparticles was noticed at 1575 and 1421 and 1597 and 1431 cm^−1^ on the Fe_3_O_4_-NPL and Fe_3_O_4_-NPF spectra.

The elongation of the peaks of the carbonyl group (aldehyde, ketone, and carboxylic acid) C = O stretch, C = C aromatic bending, C-O alkoxy stretch, C = C aromatic bending, and broad band of the alcohol/phenol O-H stretch in both spectra (Figure 3a,b) indicates good interaction of the plant extracts and the precursor. The vibration bands of green-mediated Fe_3_O_4_-NP are quite close to reported vibrations [2,31]. 

UV–visible spectroscopy was used to determine the optical property of the synthesized Fe_3_O_4_-NP and to determine their electrical conductivity (insulator, semiconductor, or conductor) through the energy band gap of the UV absorption spectra. The UV–VIS spectra for Fe_3_O_4_-NP synthesized from the leaf extract (Fe_3_O_4_-NPL) and flower extract (Fe_3_O_4_-NPF) of *C. viminalis* are shown in Figure 4 with maximum absorbance observed at 297 and 304 nm. The *C. viminalis* leaf extract and flower extract showed absorbance at 307 and 299 nm, respectively (graph not shown). The UV–VIS result for the *C. viminalis* extracts and synthesized nanoparticles (Fe_3_O_4_-NPL and Fe_3_O_4_-NPF) indicate a shift in vibrations towards shorter/higher wavelength, which suggests a good interaction of the *C. viminalis* extracts (leaf and flower) and the precursor in the formation of Fe_3_O_4_-NP [32]. The energy band gap of Fe_3_O_4_-NP synthesized from the *C. viminalis* leaf (Fe_3_O_4_-NPL) and flower (Fe_3_O_4_-NPL) extracts was obtained by Tauc’s formula, (αhv)*n* = B(hv-Eg), as 2.79 and 2.89 eV using the absorption peaks 297 and 304 nm, respectively. The values are quite close to 2.87 eV reported in the literature [33], inferring synthesized Fe_3_O_4_-NP as semiconductors since the values are greater than 5 eV. The energy band gap values of Fe_3_O_4_ are different from those of the source of the extract, which could be due to the slight absorption shift (blue and red shifts) observed for the flower extract and Fe_3_O_4_-NPF and the leaf extract and Fe_3_O_4_-NPL, resulting to quantum confinement and intrinsic effect, respectively.

The SEM (scanning electron microscopy) images of Fe_3_O_4_-NP synthesized for Fe_3_O_4_-NP synthesized by the *C. viminalis* leaf (Fe_3_O_4_-NPL) and flower (Fe_3_O_4_-NPF) extracts, respectively, revealed uniformly dispersed, agglomerated [12], and nearly spherical in shape Fe_3_O_4_-NP with rough surfaces, which suggests absorption of the carboxylate group existing in protein, acting as surfactant to adhere to the nanoparticle surface, resulting in the Fe_3_O_4_-NP stabilization as can be seen in Figure 5a,b. The mechanism of absorption is presented in Figure 6.

The energy-dispersive X-ray (EDX) analysis results revealed the presence of iron and oxygen in the Fe_3_O_4_-NP synthesized by the leaf extract (Fe_3_O_4_-NPL) with mean values of 43.68 and 56.32, respectively. The mean values of iron and oxygen for Fe_3_O_4_-NPF were 33.58 and 66.42. The reason for the differences in the weight percentages of the elemental compositions of Fe_3_O_4_-NPL and Fe_3_O_4_-NPF can be attributed to the influence of the reducing agents from each plant extract.

The TEM (transmission electron microscopy) image of Fe_3_O_4_-NP synthesized by the *C. viminalis* leaf extract (Fe_3_O_4_-NPL) shown in Figure 7ai indicate the shape of iron in Fe_3_O_4_-NPL to be nearly spherical, spatially distributed with some particles agglomerated, and the size distribution is shown in Figure 7aii using the ImageJ software. The mean diameter and average particle size for *N* = 237 of the synthesized particles (Fe_3_O_4_-NPL) were 5.313 nm and 5.3 ± 1.9. Figure 7bi shows the TEM image for Fe_3_O_4_-NPF mediated by the flower extract of *C. viminalis*. Well-dispersed spherical nanoparticles were noticed. Figure 7bii indicates the size distribution of Fe_3_O_4_-NPF with a mean diameter and average particle size of 1.127 nm and 1.1 ± 0.8 for *N* = 2163 (*N* = number of counts). 

Zeta potential analysis of Fe_3_O_4_-NP was conducted to obtain information on the surface charge of the dispersed particles (Fe_3_O_4_-NP). Particles with zeta potential values within the range of ±30 mV are widely considered stable [12,23]. The zeta potential values obtained for Fe_3_O_4_-NP synthesized from the leaf (Fe_3_O_4_-NPL) and flower (Fe_3_O_4_-NPF) extracts were −20.0 and −28.4 mV, respectively. The values are well comparable to the values reported in the literature [34]. Figure 8a,b gives a summary of the zeta potential analysis of Fe_3_O_4_-NPL and Fe_3_O_4_-NPF, respectively. The result of the zeta potential analysis showed good stability for Fe_3_O_4_-NP. A higher stability was observed for Fe_3_O_4_-NPF than Fe_3_O_4_-NPL due to greater surface charge (−28.4 mV), which could be attributed to the existence of denser electrostatic forces within the synthesized particles and compositions of the reducing agent (phenolic compounds) present in the flower extract [23,34].

TGA analysis of Fe_3_O_4_-NP was conducted with SDT Q600 V20.9 Build 20 thermogravimetric analyser of DSC-TGA standard in an unreactive nitrogen gas environment at a high furnace temperature using the RAMP method and a heating rate of 10 °C min^−1^. The thermogravimetric curves of Fe_3_O_4_-NP synthesized from the *C. viminalis* leaf (Fe_3_O_4_-NPL) and flower (Fe_3_O_4_-NPF) extracts are shown in Figure 9 with three stages of weight loss. In Fe_3_O_4_-NPL and Fe_3_O_4_-NPF, the first stage lies within 25–65 °C and 29–58 °C with an insignificant weight loss (1%), which could be attributed to loss of water and volatile solvents. The second stage started at 65 and 58 °C for Fe_3_O_4_-NPL and Fe_3_O_4_-NPF accordingly and ended at 197 °C with a weight loss of 13% and 10%, respectively, and could be assigned to loss of water due to crystallization and gas desorption. Finally, the third stage at 636 and 654 °C for Fe_3_O_4_-NPL and Fe_3_O_4_-NPF had a significant weight loss of 44% and 50%, which could be assigned to the transition phase of Fe_3_O_4_ to FeO since FeO is stable (thermodynamically) at temperatures above 570 °C [35] with a reaction formula at a possible equilibrium phase:$${\mathrm{Fe}}_{3}O_{4(s)}⇄3{\mathrm{FeO}}_{\left(s\right)}\;+\;\frac{1}{2}\;O_{2(g)}$$

The XRD spectra (Figure 10a) of synthesized Fe_3_O_4_-NP from the *C. viminalis* leaf (Fe_3_O_4_-NPL) and flower (Fe_3_O_4_-NPF) extracts conducted from 20° to 80° at 2 theta were closely similar to those reported in the literature [16,36].

The presence of a hump instead of a peak observed at 28.56° and 26.62° for Fe_3_O_4_-NPL and Fe_3_O_4_-NPF, respectively, could possibly be due to the covering of Fe_3_O_4_-NP with organic substances [35]. Therefore, the XRD pattern for the Fe_3_O_4_-NP showed deficiency in distinctive peaks, which suggests the amorphous nature of Fe_3_O_4_-NPL and Fe_3_O_4_-NPF and proves that the surface of the iron oxide is coated by organic materials from the extracts, and a similar XRD pattern was reported in the literature [36,37]. The XRD spectrum depicts Fe_3_O_4_-NP as confirmed by the colour of the precipitates formed during synthesis (Figure 10b) as opposed to the brick red/brown colour expected for Fe_2_O_3_ (haematites).

### 3.5. Electrochemical Characterization of Fe_3_O_4_-NP 

Electrochemical comparative studies on iron (III) oxide nanoparticles synthesized from *Callistemon viminalis* leaf and flower extracts (Fe_3_O_4_-NPL and Fe_3_O_4_-NPF) were carried out using cyclic voltammetry in 5 mM potassium hexacyanoferrate (III) (K_3_[Fe(CN)_6_]) solution prepared in 0.1 M phosphate buffer solution (PBS) as a supporting electrolyte at 25 mVs^−1^ scan rate in order to examine the electrocatalytic performance of the synthesized iron (III) oxide nanoparticles. The cyclic voltammogram of the electrochemical experiment is shown in Figure 11 for the bare SPCE (screen-printed carbon electrode) and modified SPCEs with Fe_3_O_4_-NPL and Fe_3_O_4_-NPF with anodic potential peaks found in SPCE-Fe_3_O_4_-NPL 0.2591, 0.2488, 0.2557 V, respectively, for the probe (K_3_Fe(CN)_6_), while the sequence of the oxidation (anodic) current peaks at the bare SPCE and modified electrodes is SPCE-Fe_3_O_4_-NPL (82.127 μA) > SPCE-Fe_3_O_4_-NPF (26.010 μA) > SPCE (12.820 μA). The SPCE-modified electrodes displayed higher anodic current (Ipa) response than the bare SPCE, which proved a successful modification of the bare screen-printed carbon electrode. The Ipa for the SPCE-Fe_3_O_4_-NPL was found to be six times higher than that for the bare SPCE and three times higher than that for SPCE-Fe_3_O_4_-NPF, suggesting SPCE-Fe_3_O_4_-NPL to be more electrically conductive than SPCE-Fe_3_O_4_-NPF, suggesting excellent ionic interaction among the nanoparticles’ large surface area, which in turn allowed free flow of ions/electrolytes in and out of the electrode surface. The high diffusivity of ions at the same electrode is ascribed to higher surface area for effective diffusion. SPCE-Fe_3_O_4_-NPF was two times higher than the bare SPCE. The large surface-to-volume ratio and high electrical conductivity of iron (III) oxide nanoparticles (Fe_3_O_4_-NP) were responsible for the high electron transport at the modified electrodes (SPCE-Fe_3_O_4_-NPL and SPCE-Fe_3_O_4_-NPF). The ratio of the anodic-to-cathodic peak currents was greater than 1 and the unity for SPCE-Fe_3_O_4_-NPL and SPCE-Fe_3_O_4_-NPF, suggesting an irreversible and reversible redox process accordingly.

The influence of varying scan rates on the modified electrodes was determined for further understanding of the kinetics of the electron transport process at the electrodes (SPCE-Fe_3_O_4_-NPL and SPCE-Fe_3_O_4_-NPF) using cyclic voltammetry in PBS containing 5 mM K_3_Fe(CN)_6_ solution. Increase in scan rates resulted in an increased anodic current response as seen in Figure 12ai,bi. Regression values of 0.9968 and 0.9969 for the respective modified electrodes were obtained from the plot of current (Ipa) against the square root of the scan rate (Figure 12aii,bii), which confirmed a diffusion-controlled process at the modified SPCE electrodes. 

The parameters used to compare the electrochemical properties of the modified electrodes were electron transport coefficients (α), which were calculated to be 0.53 and 0.56 for the SPCE-Fe_3_O_4_-NPL and SPCE-Fe_3_O_4_-NPF electrodes from the slope values of the linear plots of Ep (peak potentials) versus the logarithm of scan rates (log mVs^−1^) (graph not shown) equal to $\frac{-2.3RT}{\propto nF}$ and $\frac{2.3RT}{\left(1-\propto \right)nF}$ assigned to cathodic and anodic peaks, respectively, by Lavarion [38], where Ep is the slope, *R* represents the gas constant (8.3142 JK^−1^ mol^−1^), and *T* is room temperature in K (298). The number of electrons transferred (*n*) at the respective electrodes was one for both electrodes. The values of the slope from the plots of the respective modified electrodes were found to be 0.11235 and 0.13916 Vdec^−1^. The Tafel value “b” was obtained using Equation (2).
(2)$$\mathrm{Ep}=\frac{b}{2}\left(\log\;(mVS{-}^{1}\right)+constant$$

The values of b for the modified electrodes SPCE-Fe_3_O_4_-NPL and SPCE-Fe_3_O_4_-NPF were 0.224 and 0.278 Vdec^−1^, respectively. The Tafel value obtained for both electrodes were found higher than the theoretical value (0.118 Vdec^−1^) for one-electron transport. The differences in the value of “b” in theory and practice suggest surface assimilation of reactants or intermediates on the electrode surfaces [39]. From the slope of the peak current (*Ip*) versus the scan rate, the diffusion coefficient (*D*) was calculated to be 3.178 × 10^−6^ and 1.0714 × 10^−6^ cm^2^ s^−1^ for SPCE-Fe_3_O_4_-NPL and SPCE-Fe_3_O_4_-NPF, respectively, using Randles–Sevcik Equation (3).
(3)$$I_{p}=(2.69\;\times \;10^{5})n^{\frac{3}{2}}AD^{\frac{1}{2}}C\;v^{\frac{1}{2}}$$

In the *Ip* maximum current (amperes), *A* is the electrode area of SPCE (0.1257 cm^2^), *C* is the concentration of the bulk solution (mol/cm^3^), and v is the scan rate in *Vs*^−1^. The surface concentrations of the respective modified electrodes (SPCE-Fe_3_O_4_-NPL and SPCE-Fe_3_O_4_-NPF) were calculated to be 3.47 and 1.91 μmol/cm^2^ by applying Equation (4).
(4)$$I_{p}\;=\;\frac{n^{2}F^{2}A\Gamma^{*}v}{4RT}$$

The greater surface concentration value obtained for SPCE-Fe_3_O_4_-NPL as compared with that obtained for the SPCE-Fe_3_O_4_-NPF-modified electrode confirmed the reason for the high electrical conductivity of the electrode. 

A stability study of modified electrodes SPCE-Fe_3_O_4_-NPL and SPCE-Fe_3_O_4_-NPF was examined in PBS with pH 7.4 containing 5 mM K_3_Fe(CN)_6_ solution using cyclic voltammetry for 12 repetitive scans. The anodic current drops observed for the SPCE-Fe_3_O_4_-NPL and SPCE-Fe_3_O_4_-NPF electrodes were 12% and 9%, respectively, considering the first and last scans of the electrodes (graph not shown). Comparing the stability of the two electrodes, SPCE-Fe_3_O_4_-NPF was more stable than SPCE-Fe_3_O_4_-NPL, suggesting lesser etching of the SPCE-Fe_3_O_4_-NPF electrode surface caused by the existence of cyanide ions (CN^−^) discharged from the probe solution (K_3_Fe(CN)_6_) [40]. 

### 3.6. Antimicrobial Activity of Green-Mediated Fe_3_O_4_-NP on Selected Bacterial Pathogens 

Table 1 presents the zones of inhibition of green-mediated Fe_3_O_4_-NP activity against selected bacterial pathogens (microbes) exposed to varying concentrations (100 and 150 µg/mL) of nanoparticles. The antimicrobial activity of Fe_3_O_4_-NPL against *Bacillus cereus* ranged from 8 to 13 mm and was highest in Fe_3_O_4_-NPF at 150 µg/mL. However, there was no significant difference between the antimicrobial activities of Fe_3_O_4_-NPL and Fe_3_O_4_-NPF. A similar trend was observed in the antimicrobial activities of nanoparticles against *E. coli*, *Enterococcus faecalis*, *Enterococcus faecium*, *Salmonella enteritidis*, and *Vibrio cholerae*. At 150 µg/mL, the antimicrobial activity of Fe_3_O_4_-NPF (14 mm) was highest against the tested pathogens except in *Salmonella typhimurium* (8 mm). In *Vibrio cholerae*, it could be suggested that the antimicrobial activity of Fe_3_O_4_-NP was influenced not only by the concentration of nanoparticles but also by the part of the plant used and the phytochemical present because higher zones of inhibition were obtained in Fe_3_O_4_-NP synthesized by the flower extract compared with those obtained in Fe_3_O_4_-NP synthesized by the leaf extract in most cases, which could be attributed to high phenolic content in the flower extract, leading to an improved reducing ability of nanoparticles. Differences in bacterial cell structures also play a role in the inhibition of microorganisms [41], which could be the case of high antimicrobial activity of Fe_3_O_4_-NPL against *Salmonella typhimurium* compared with that of Fe_3_O_4_-NPF. The structure of the bacterial cell wall and the nanoparticle concentration influence the ability of nanoparticles to alter the replication process in bacteria and the toxic activities of nanoparticles against pathogens [41]. Nanoparticles could generate a pit within the cell wall, resulting to an improved permeability, which could lead to the death of bacterial cells [39]. Nanoparticles have the ability to switch off the activity of cellular enzymes and DNA through the electron-donating group (thiols, carboxylates, amides, imidazoles, indoles, and hydroxyls) [42]. This electron-donating group could be from the phytochemical components of the plant used in the green synthesis of Fe_3_O_4_-NP. Previous studies have reported the activity of nanoparticles against pathogenic and drug-resistant bacteria [43]. The result shown in Table 1 reveal that *Vibrio cholerae* was more susceptible to Fe_3_O_4_-NPF, followed by *Salmonella enteritidis*, *Enterococcus faecalis*, and *E. coli*, and from leaves, *Vibrio cholerae*, *Salmonella typhimurium*, *Enterococcus faecium*, *E. coli*, and *Enterococcus faecalis*. The inhibition zones of selected bacterial pathogens (microbes) increased with increase in the concentration of Fe_3_O_4_-NP in µg/mL.

## 4. Conclusions

This work reports a comparative study of Fe_3_O_4_-NP (Fe_3_O_4_-NPL and Fe_3_O_4_-NPF) synthesized by using *C. viminalis* leaf and flower extracts typical in the North West province of South Africa. The extracts acted as a reducing agent, reducing iron (III) ions (Fe^3+^) to zero valence metal. Energy band gap calculation through UV–visible spectra revealed synthesized nanoparticles to be semiconductors. FTIR spectra showed functional groups such as phenols and ketones to be present in plant extracts, which are believed to be responsible in the formation of synthesized iron (III) oxide nanoparticles. SEM micrographs revealed rough surfaces of synthesized nanoparticles, confirming the absorption of extracts on the surface of the nanoparticles. TEM analysis revealed the shape of the synthesized particles to be spherical, especially Fe_3_O_4_-NPF. Zeta potential analysis revealed negative surface charges of the synthesized nanoparticles and their electrical stability with Fe_3_O_4_-NPF being more stable due to its high surface charge, which could be attributed to the existence of denser electrostatic forces within the synthesized particles and compositions of the reducing agent (flower extracts). Additionally, antimicrobial studies conducted on synthesized nanoparticles against selected pathogens (microbes) revealed good antimicrobial activity for both Fe_3_O_4_-NPL and Fe_3_O_4_-NPF. Electrochemical studies at a modified screen-printed electrode confirmed better electrochemical activity at a modified SPCE of Fe_3_O_4_-NPL than Fe_3_O_4_-NPF considering the current (Ipa) response, although better stability was noticed at SPCE/Fe_3_O_4_-NPF. However, both modified Fe_3_O_4_-NPL and Fe_3_O_4_-NPF can be good catalysts for fabricating electrochemical sensors for biological and environmental analytes.

## Figures and Tables

**Figure 1 materials-13-04894-f001:**
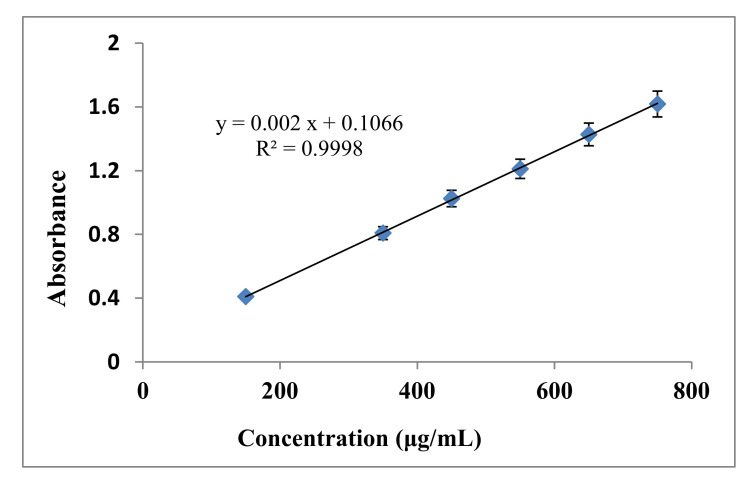
Gallic acid standard calibration curve.

**Figure 2 materials-13-04894-f002:**
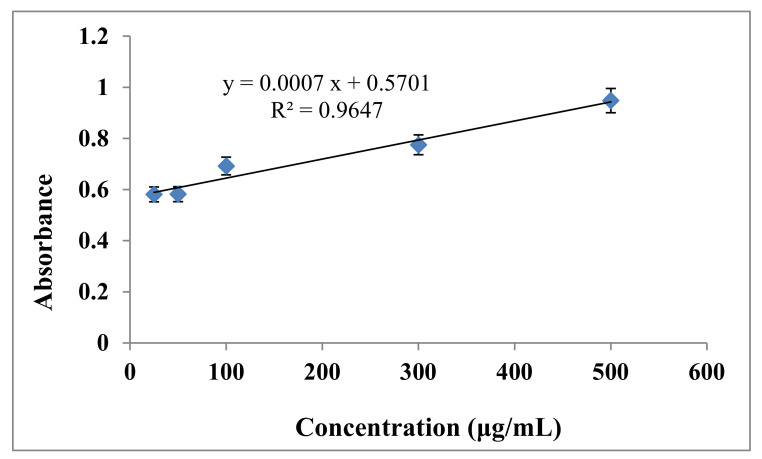
Quercetin standard calibration curve.

**Figure 3 materials-13-04894-f003:**
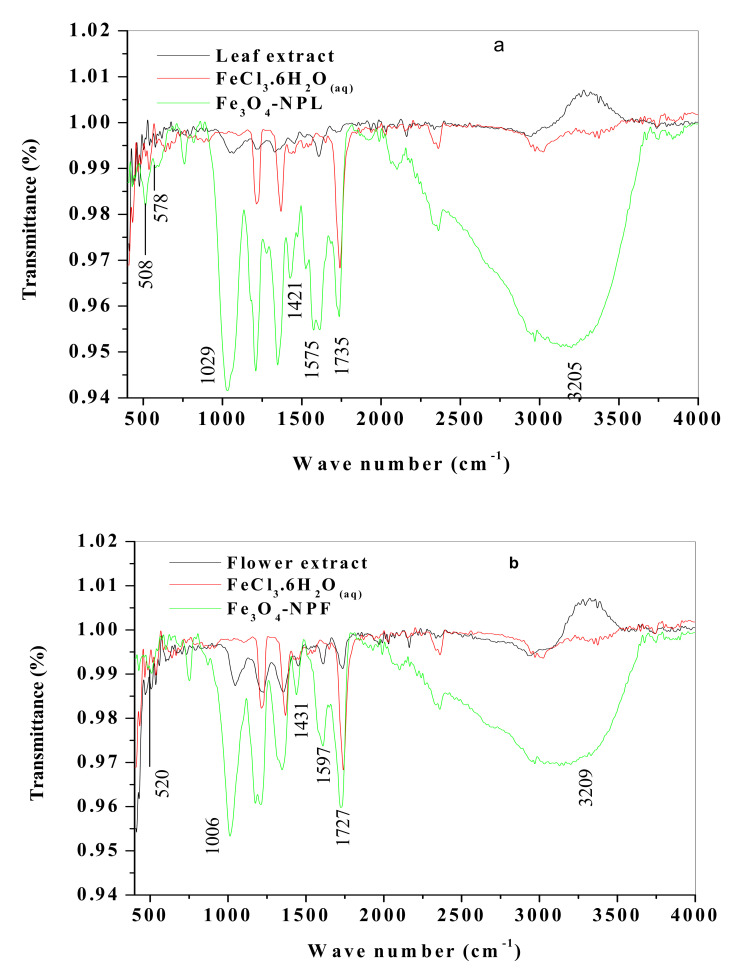
FTIR spectra of Fe_3_O_4_-NP synthesized by *C. viminalis* (**a**) leaf and (**b**) flower extracts, respectively.

**Figure 4 materials-13-04894-f004:**
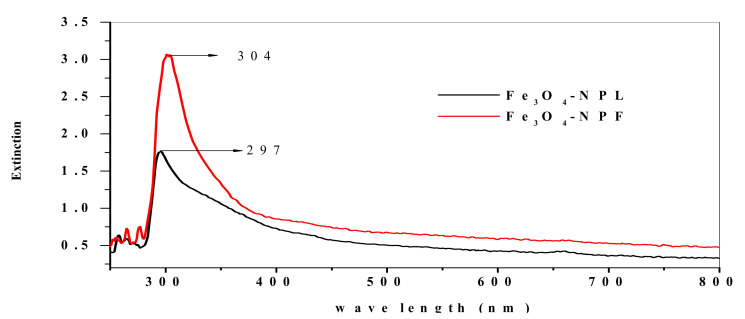
UV–visible spectra of Fe_3_O_4_-NPL and Fe_3_O_4_-NPF synthesized by leaf and flower extracts, respectively.

**Figure 5 materials-13-04894-f005:**
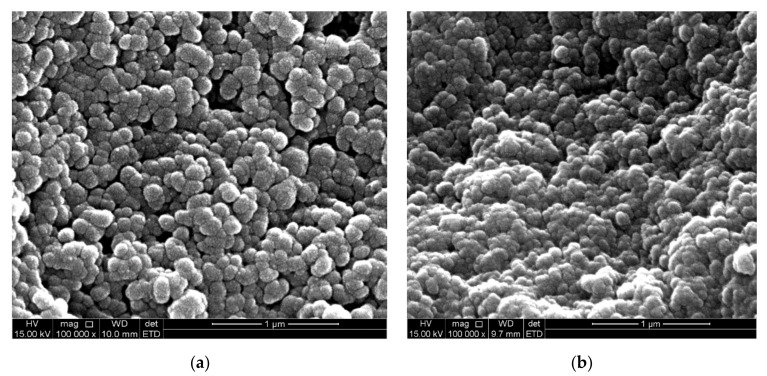
Scanning electron microscopic image of (**a**) Fe_3_O_4_-NPL and (**b**) Fe_3_O_4_-NPF.

**Figure 6 materials-13-04894-f006:**
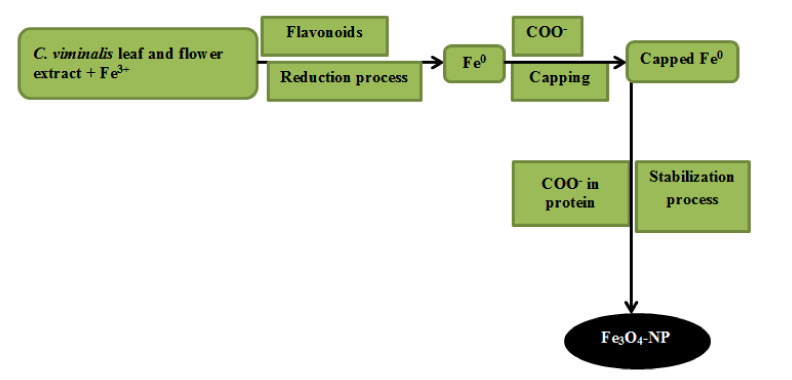
Possible absorption mechanism of bio-reduced Fe_3_O_4_-NP.

**Figure 7 materials-13-04894-f007:**
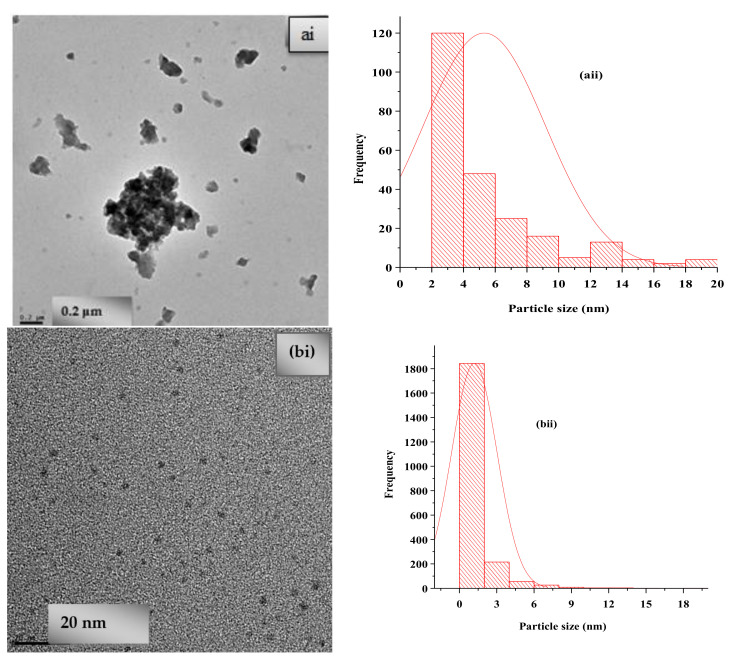
TEM images of (**ai**) Fe_3_O_i_-NPL and (**bi**) Fe_3_O_4_-NPF and particle size distribution of (**aii**) Fe_3_O_i_-NPL and (**bii**) Fe_3_O_4_-NPF.

**Figure 8 materials-13-04894-f008:**
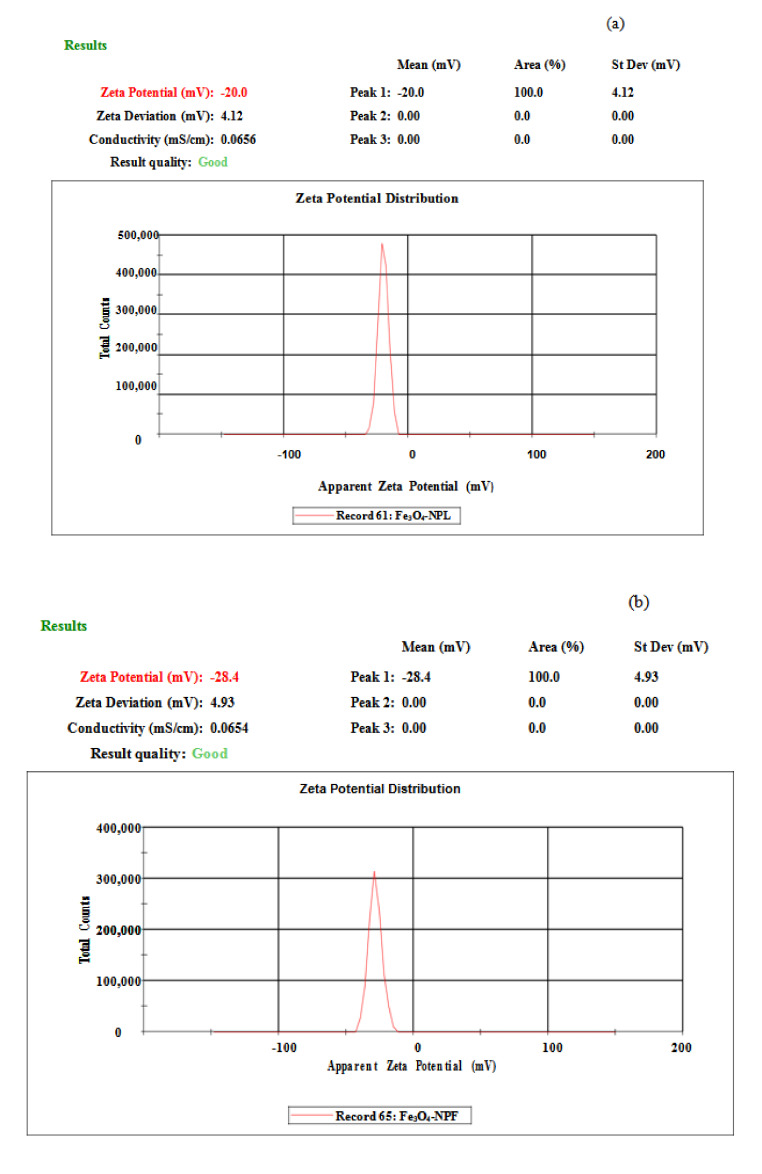
Zeta potential of (**a**) Fe_3_O_4_-NPL and (**b**) Fe_3_O_4_-NPF.

**Figure 9 materials-13-04894-f009:**
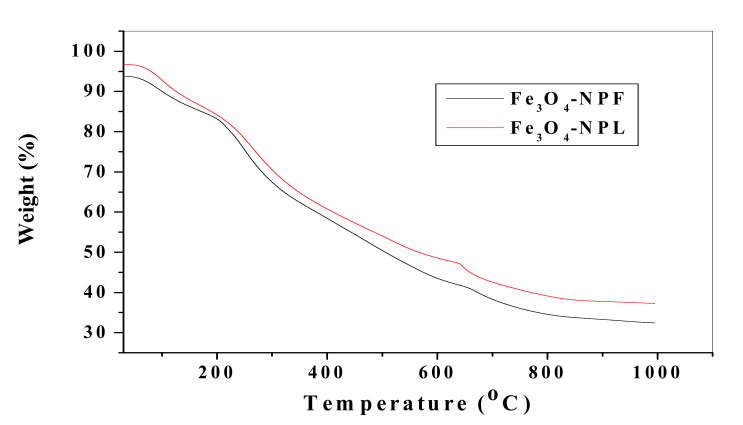
Thermogravimetric curves of Fe_3_O_4_-NPs from *C. viminalis* leaf and flower extracts.

**Figure 10 materials-13-04894-f010:**
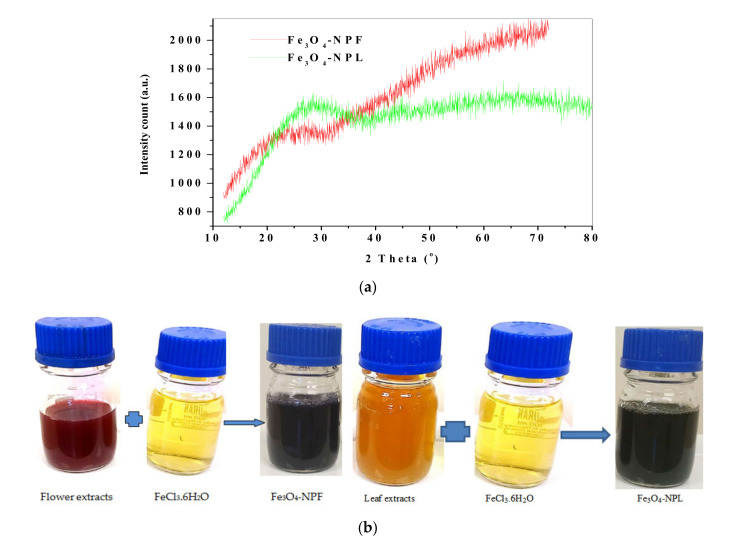
(**a**) X-ray diffraction pattern of Fe_3_O_4_-NP from *C. viminalis* leaf and flower extracts, (**b**) colour formation of Fe_3_O_4_-NPF and Fe_3_O_4_-NPL with FeCl_3_. 6H_2_O during synthesis.

**Figure 11 materials-13-04894-f011:**
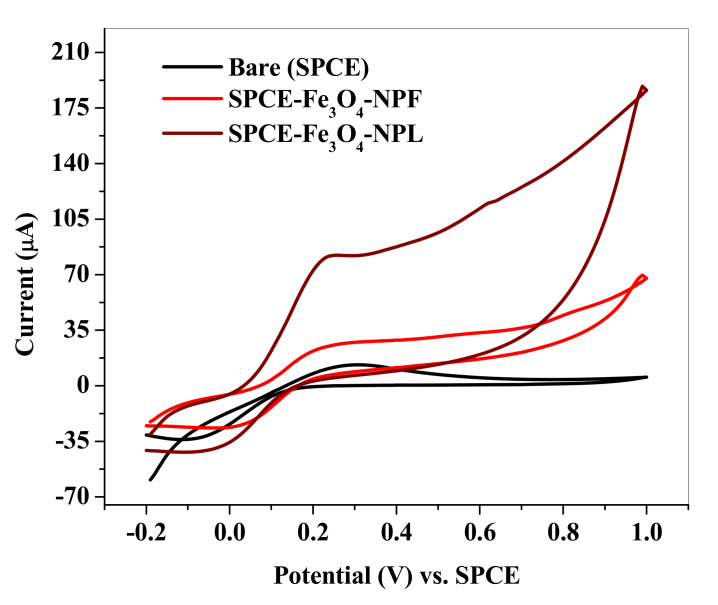
Cyclic voltammogram of bare SPCE, SPCE-Fe_3_O_4_-NPF, and SPCE-Fe_3_O_4_-NPL at 25 mVs^−1^ scan rate in 0.1 M PBS (pH 7.4) containing 5 mM K_3_Fe(CN)_6_ solution.

**Figure 12 materials-13-04894-f012:**
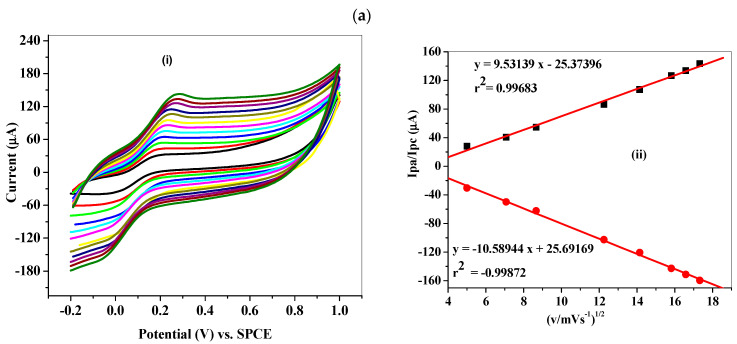
Cyclic voltammogram obtained for (**a**(**i**)) SPCE/Fe_3_O_4_-NPL and (**b**(**i**)) SPCE/Fe_3_O_4_-NPF in PBS pH 7.4 containing 5 mM K_3_Fe(CN)_6_ solution (scan rate: 25–300 mVs^−1^; inner to outer) and linear plots of Ipa/Ipc vs. square root (**a**(**ii**)) and (**b**(**ii**)) of scan rate for SPCE-Fe_3_O_4_-NPL and SPCE-Fe_3_O_4_-NPF, respectively.

**Table 1 materials-13-04894-t001:** Antimicrobial activity of Fe_3_O_4_-NP on selected microbes.

Pathogens	100 µg/mL of Fe_3_O_4_-NP Synthesized from Leaves	150 µg/mL Fe_3_O_4_-NP Synthesized from Leaves	100 µg/mL vFe_3_O_4_-NP Synthesized from Flowers	150 µg/mL Fe_3_O_4_-NP Synthesized from Flowers	Ciprofloxacin * (mm)
*Bacillus cereus* (ATCC 10876)	8	12	10	13	30
*E. coli* (ATCC 25922)	11	13	11	14	40
*Enterococcus faecalis* (ATCC 29212)	11	13	12	14	37
*Enterococcus gallinarum* (ATCC 700425)	7	9	12	11	28
*Enterococcus faecium* (ATCC 701221)	8	14	10	12	38
*Salmonella typhimurium* (ATCC 14028)	12	14	9	8	31
*Salmonella enteritidis* (ATCC 13076)	10	13	12	15	35
*Vibrio cholerae*	13	14	14	25	40

* Ciprofloxacin served as the positive control.

## Data Availability

The data that support the findings of this study are available from the corresponding author upon reasonable request.

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
