# Peer review of "Synthesis, Electrochemical Studies, and Antimicrobial Properties of Fe3O4 Nanoparticles from Callistemon viminalis Plant Extracts"

_materials, 2020, doi:10.3390/ma13214894_

Round 1

Reviewer 1 Report

Please find the attached document for the comments. 

Reviewer 2 Report

Please show attached file.

Round 2

Round 3

Reviewer 2 Report

The sentences from L353 to 357 is complicated and too long to understand. It became worse than before the correction by making two sentences into one sentence with deleting some informations.

"The Ipa for the SPCE-Fe3O4-NPL was found to be six times higher than the bare SPCE and three times higher than SPCE-Fe3O4-NPF suggesting SPCE-Fe3O4-NPL to be more electrically conductive than SPCE-Fe3O4-NPF suggesting excellent ionic interaction among the nanoparticles, large surface area which in turn allowed free flow of ions/electrolyte in and out of the electrode surface."

It is better to be recorrected.